# Evaluation of skin doses for cone-beam computed tomography in dentomaxillofacial imaging: A preclinical study

**Carolin Sophie Reidelbach**[1]*, **Jakob Neubauer**[1], **Maximilian Frederik Russe**[1], **Jan Kusterer**[2], **Wiebke Semper-Hogg**[2]

**1** Department of Diagnostic and Interventional Radiology, Medical Center–University of Freiburg, Faculty of Medicine, University of Freiburg, Freiburg, Germany, **2** Department of Oral and Craniomaxillofacial Surgery, Center for Dental Medicine, Medical Center—University of Freiburg, Faculty of Medicine, University of Freiburg, Freiburg, Germany

* carolin.reidelbach@uniklinik-freiburg.de

## Abstract

### Purpose

Evaluation of skin organ doses in six different cone-beam computed tomography scanners (CBCT) dedicated to dentomaxillofacial imaging. Our hypothesis is that the dose varies between different devices, protocols and skin areas.

### Materials and methods

An anthropomorphic adult head and neck phantom was used to which a dosimeter (Water-proof Farmer® Chamber, PTW, Freiburg, Germany) was attached to anatomic landmarks of both parotid glands, both ocular lenses, the thyroid gland and the neurocranium. CBCT examinations were performed on six different CBCT devices dedicated to dentomaxillofacial imaging with standard settings and, if available, also in high dose settings. Measurements were repeated five times each.

### Results

The measured mean skin doses ranged from 0.48 to 2.21 mGy. The comparison of the region based dose evaluation showed a high correlation between the single measurements. Furthermore, the distribution of doses between regions was similar in all devices, except that four devices showed side differences for the dose of the parotid region and one device showed side differences for the lens region. The directly exposed regions, such as the parotid glands, showed significant higher values than the more distant regions like the neurocranium.

When comparing examination protocols, a significant difference between the standard dose and the high dose acquisitions could be detected. But also a significant dose difference between the different CBCTs could be shown. 3D Accuitomo 170 (Morita, Osaka, Japan) showed the highest absorbed mean dose value for standard settings with 2.21 mGy,

**Data Availability Statement:** All relevant data are within the paper and its Supporting Information files.

**Funding:** The study was partly funded by Dürr Dental SE: partial purchase oft he phantom head, no influence on collection, analysis and interpretation of data, writing of the paper and/or decision to submit for publication. A grant number does not exist. There was no additional external funding received for this study.

**Competing interests:** Wiebke Semper-Hogg was a consultant of Dürr Dental SE. https://www.duerrdental.com/en/home/. This does not alter our adherence to PLOS ONE policies on sharing data and materials.

especially at the directly exposed regions and their adjacent organs. The lowest mean value for standard settings was achieved with VGi evo (NewTom, Verona, Italy) with 0.48 mGy.

## Conclusion

Repeated measurements of skin organ doses in six different CBCT scanners using a surface dosimeter showed side differences in distribution of dose in five devices for the parotid and lens region. Additionally, significant dose differences between the devices could be detected. Further studies should be performed to confirm these results.

## Introduction

Cone-beam computed tomography (CBCT) is a modality more and more on the rise in dento-maxillofacial imaging [1]. Due to its three-dimensional, superimposition-free presentation, it is now often used instead of conventional radiography [2, 3]. Its simple construction and easy handling make it accessible not only for hospitals, but also for dental practices. However, the main drawback of cross sectional imaging is the higher radiation dose compared to radiography, even though several studies showed a lower radiation dose for CBCT than for conventional multidetector computed tomography (MDCT) [1, 2].This implies that many radiation sensitive organs such as the salivary glands are exposed to the x-ray beam without being depicted and thus with no diagnostic value. Therefore, several studies examined the distribution of organ and effective doses in CBCT, showing that the salivary glands, the remainder tissue and the thyroid are the most exposed organs [4–6]. With numerous manufacturers and devices on the market, CBCT scanners differ significantly in their buildup and their settings such as fields of view (FOV), setting of the isocenter, positioning of the rotational arc and dose protocols, leading to differences in amount and distribution of dose and making it difficult to draw comparisons between the devices and, furthermore, to define standard exposition values that apply for all devices [7]. In previous studies, the radiation exposure of different CBCTs was examined using thermoluminescent dosimeters (TLDs) or metal oxide semiconductor field effect transistors (MOSFETs) for determination of dose, and both seem to be an equivalent method for this application [8, 9]. Since only few studies refer to side differences regarding dose exposition in CBCT-imaging [7], the purpose of this study was to investigate the significant dose differences between the left and the right sided organs and the differences among six CBCT devices.

## Materials and methods

### Head and neck phantom

As study object we used an anthropomorphic adult head and neck phantom representing the shape of an adult human (CBCT QA and Dosimetry 2690201, CIRS, Virginia, USA). This phantom was custom-made with a borehole in the middle of the phantom´s head for intracranial positioning of the measuring chamber.

### Dose measurements

For dose measurements, we used a 3D measuring chamber dosimeter due to its simple applicability (Waterproof Farmer® Chamber, PTW, Freiburg, Germany). The dosimeter was attached to the anatomic landmarks of both parotid glands, both ocular lenses, and the thyroid gland, as these organs are known to be the most radiation sensitive organs in the head and

neck area. Measurements for the neurocranium were performed by placing the dosimeter in the borehole in the middle of the phantom. Dose measurements were repeated five times each, the dosimeter was positioned anew each time.

### CBCT devices

CBCT examinations were performed with six different devices in standard and high dose settings, if available, to reproduce the large variety of the scanners used in clinical and practice settings: VistaVox S (Dürr Dental, Bietigheim-Bissingen, Germany), Orthophos SL 3D (Sirona, Wals, Austria), 3D Accuitomo 170 (Morita, Osaka, Japan), VGi evo (NewTom, Verona, Italy), ProMax 3D Mid (Planmeca, Helsinki, Finland), CS 9300 (Carestream, Atlanta, GA, USA). Dose and FOV settings are shown in Table 1.

### Statistics

For a descriptive analysis, mean value, median value and standard deviation were calculated. Linear mixed models were used to compare the regions within a device class and the values of different devices per region to examine statistically significant dose differences concerning the left- and the right-sided organs and differences between the six CBCT devices. Furthermore, pairwise comparisons were performed. Therefore, Scheffe´s method was applied to correct for the multiple testing problem and adjustment of p-values, respectively. A p-value less than 0,05 was set to be statistically significant. The statistical analysis was performed with the software STATA (StataCorp LLC, College Station, TX, USA) [10].

## Results

The comparison of the region-based dose evaluation showed a high correlation between the single measurements. The measured mean skin doses ranged from 0.48 to 2.21 mGy. Morita 3D Accuitomo 170 showed the highest absorbed mean dose value for standard settings with 2.21 mGy. The lowest absorbed mean dose value for standard settings was achieved by New-Tom VGi evo with 0.48 mGy. When comparing examination protocols, a significant difference between the standard dose and the high dose acquisitions could be detected as expected. Mean organ skin doses for standard and high dose settings for each device are shown in Table 2.

The highest organ dose for standard settings was achieved with Morita 3D Accuitomo 170 for the right parotid gland with 4.51 mSv, whilst the lowest organ dose for standard settings

**Table 1. Dose and FOV settings for all six devices for standard dose (SD) and, if available, high dose (HD) settings.**

|  | FOV | Standard dose (SD) settings | High dose (HD) settings |
|---|---|---|---|
| **VistaVox S** | 100 (130) x 85 | 79 kV | 94 kV |
|  |  | 5.0 mA | 9.0 mA |
| **Orthophos SL 3D** | 110 x 100 | 85 kV | 85 kV |
|  |  | 6.0 mA | 10 mA |
| **3D Accuitomo 170** | 100 x 100 | 90 kV | - |
|  |  | 7.5 mA |  |
| **VGi evo** | 100 x 100 | 110 kV | 110 kV |
|  |  | 3.0 mA | 7.5 mA |
| **ProMax 3D Mid** | 100 x 100 | 90 kV | 90 kV |
|  |  | 6.3 mA | 8.0 mA |
| **CS 9300** | 100 x 100 | 90 kV | - |
|  |  | 4.0 mA |  |

**Table 2. Mean organ skin doses for standard dose (SD) and high dose settings (HD), if available, for each device.**

|  | Mean organ skin dose (SD) | Mean organ skin dose (HD) |
|---|---|---|
| **VistaVox S** | 0.64 mSv | 1.74 mSv |
| **3D Accuitomo 170** | 2.21 mSv | - |
| **VGi evo** | 0.48 mSv | 1.38 mSv |
| **Orthophos SL 3D** | 1.17 mSv | 2.51 mSv |
| **ProMax 3D Mid** | 0.85 mSv | 1.34 mSv |
| **CS 9300** | 0.94 mSv | - |

was achieved with Dürr VistaVox S for the neurocranium with 0.064 mSv. The directly exposed organs, in the first place the parotid glands, showed significantly higher organ doses than the more distant regions like the neurocranium and the thyroid. Distribution of dose for standard settings broken down by regions for each device is shown in Fig 1.

Pairwise comparison with Scheffe´s method showed statistically significant side differences with $p < 0.05$ for skin organ doses for standard dose settings in four devices for the parotid region (Dürr VistaVox S, New Tom VGi evo, Sirona Orthophos SL 3D, Carestream CS 9300) and in one device for the lens region (Planmeca ProMax 3D Mid). For high dose settings, statistically significant side differences with $p < 0.05$ for skin organ doses could be detected in three devices for the parotid region (Dürr VistaVox S, New Tom VGi evo, Sirona Orthophos SL 3D) and in two devices for the lens region (Sirona Orthophos SL 3D, Planmeca Promax 3D Mid).

Pairwise comparison with Scheffe´s method showed statistically significant differences with $p < 0.05$ for skin organ doses between the devices in the region-based comparisons.

## Discussion

Due to its three-dimensional, superimposition-free presentation, CBCT is an imaging method more and more on the rise in dentomaxillofacial imaging, often replacing radiography in

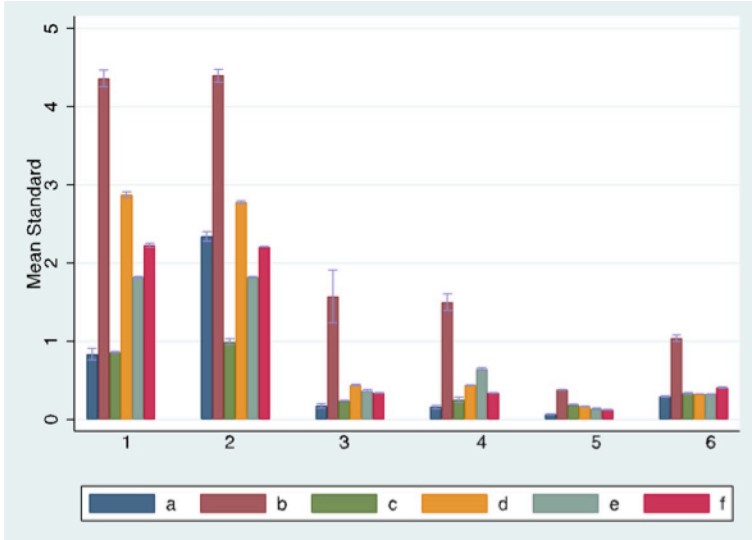

**Fig 1. Mean organ doses for standard settings.** Mean organ doses for standard settings for each region (1 = left parotid, 2 = right parotid, 3 = left lens, 4 = right lens, 5 = intracranial, 6 = thyroid) and device (a = Dürr VistaVox S, b = Morita 3D Accuitomo 170, c = New Tom VGi evo, d = Sirona Orthophos SL 3D, e = Planmeca ProMax 3D Mid, f = Carestream CS 9300).

clinical and practice settings. Despite the advantages, the importance of radiaton protection must be kept in mind, especially since radiation sensitive organs such as the thyroid region and the salivary glands are located in the irradiated area. Since there are applicational and structural differences between the devices in the market, dose differences between the scanners and differences in distribution of dose must be assumed. The aim of this study was to examine dose differences between six CBCT scanners and to examine side differences in distribution of dose in order to improve the diagnostic work-up with regard to lower the radiation induced cancer risk in dentomaxillofacial CBCT imaging in the long run.

In this preclinical study on six CBCT devices dedicated to dentomaxillofacial imaging, we demonstrate that repeated measurements of skin organ doses using a surface dosimeter show side differences in distribution of dose in four devices for the parotid region and in one device for the lens region. Additionally, in the comparison of the scanners, significant dose differences could be detected.

The only device not showing side differences in standard or high dose settings was Morita 3D Accuitomo 170, which rotates in a 360˚ mode. On the other hand, this device achieved the highest mean organ dose for standard settings, which might be caused by its full rotation mode. There are serveral studies examining dose effects of 180˚ versus 360˚ rotation angles, showing that 180˚ rotation modes are causing lower effective doses [11]. The side differences in distribution of dose for the parotid and lens region might partly be explainable by the fact that some devices do not fully rotate in an 360˚ arc. However, in our study, also another device with a full rotation mode (New Tom VGi evo) shows side differences, which leads to the conclusion that other factors must contribute to side differences as well.

Since the FOV and the dose protocols were defined by the manufacturer, they could only be adapted approximately between the devices for the experimental set-up. According to Pauwels et al., it is possible that this and other default parameters such as the setting of the isocenter and the positioning of the rotational arc may lead to side differences [12]. Also, the detected dose differences between the scanners clearly point in this direction. Furthermore, for two devices, high dose settings were not available. But as dose distribution between standard and high dose protocols was nearly similar for the devices with both settings available, new findings for dose distribution may not to be expected.

Additionally, dose saving techniques, where the x-ray beam is turned off intermittently or the mAs is modulated during the rotation, could have contributed to side differences as well, leading to a different distribution of the x-ray spectrum [13]. In literature, only few studies have examined side differences in dentomaxillofacial imaging, especially for the parotid and lens region, describing that the size and position of the FOV and the rotation arc may lead to dose asymmetry [7]. There are studies in which dose measurements were performed on both sides of paired organs, but side differences have not been discussed [14]. Especially in dentistry, where CBCT is a rising frequently used imaging method, our results according side differences may improve the diagnostic work-up in clinical and practice settings in order to lower the radiation exposure [15].

As the parotid glands are lying in the irradiated area, it is obvious that the skin adjacent to them receives the highest mean organ doses. These findings go in line with those of other studies, in which the parotid glands and the remainder tissue are the most exposed organs [16, 17]. It can be assumed that the less frequent side differences for the lens region might result from that the lenses are not in the primary beam path and therefore the overall radiation exposure is lower than for the skin adjacent to the parotid glands. The radiation exposure for the lens region must be predominantly caused by scattered radiation, which is distributed more diffusely. These results go in line with other studies, in which the lens region received lower radiation values [6, 16, 17]. This could explain why the side differences in radiation exposure

occurred less frequently in this region than in the parotid region. In this respect, and since the thyroid gland and the neurocranium are not in the primary beam path, they receive much lower skin organ doses.

Our data refer to skin organ dose measurements on the surface of a head and neck phantom, only the intracranial dose was measured by a dosimeter placed in a borehole in the phantom, the other dose values have to be considered as approximated values. As a limitation, this suggests that the real organ doses can be lower than the doses measured in this study. The design was chosen because of its simple repeatability and because of using a surface dosimeter. Nevertheless, regarding only the dose distribution, our data go in line with other studies, in that the directly exposed organs, especially the parotid glands, showed significantly higher organ doses than the more distant regions like the neurocranium and the thyroid [6, 18]. Additionally, according to dose range, our data are consistent with other studies examining skin doses, in that dose ranges from 0.04 to 4.62 mGy could be found [19, 20].

According to our knowledge, this is the first study using a 3D measuring chamber on this problem. There are several studies comparing TLDs and MOSFETs for determination of dose, arriving at the conclusion that both seem to be a feasible and nonetheless equivalent method for this issue [8, 18]. This measuring chamber has not yet been tested for this application. However, since previous research results are in line with our measurement data, the measurement chamber seems to be a comparable measurement method. Placing the dosimeter anew each of the five measurements may lead to varied, possibly inclined positioning of the measuring chamber and consequently to erroneous measurements. But since there was a high correlation between the single measurements and a similar distribution of dose compared to other studies, this can be considered insignificant.

In summary, the possibility of asymmetric dose distribution should be considered in the assessment of and in future studies on radiation dose in dentomaxillofacial CBCT. Furthermore, significant dose differences between the individual devices must be expected. Integrating this knowledge in the diagnostic work-up in clinical practice could lower radiation exposure and radiation induced cancer risk. Further studies should be performed to confirm these preliminary results.

## Supporting information

**S1 Data.**
(XLSX)

**S1 File.**
(DOCX)

## Author Contributions

**Conceptualization:** Wiebke Semper-Hogg.

**Data curation:** Jan Kusterer, Wiebke Semper-Hogg.

**Formal analysis:** Maximilian Frederik Russe, Jan Kusterer, Wiebke Semper-Hogg.

**Funding acquisition:** Carolin Sophie Reidelbach, Wiebke Semper-Hogg.

**Investigation:** Carolin Sophie Reidelbach.

**Methodology:** Jan Kusterer, Wiebke Semper-Hogg.

**Project administration:** Wiebke Semper-Hogg.

**Supervision:** Jakob Neubauer, Maximilian Frederik Russe, Wiebke Semper-Hogg.

**Visualization:** Wiebke Semper-Hogg.

**Writing – original draft:** Carolin Sophie Reidelbach.

**Writing – review & editing:** Jakob Neubauer, Maximilian Frederik Russe, Jan Kusterer, Wiebke Semper-Hogg.

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
