## [Decision Letter · Decision Letter 0]

10 Dec 2020

PONE-D-20-34852

Evaluation of skin doses for Cone-Beam Computed Tomography in Dentomaxillofacial Imaging - A preclinical study.

PLOS ONE

Dear Dr. Reidelbach,

Thank you for submitting your manuscript to PLOS ONE. After careful consideration, we feel that it has merit but does not fully meet PLOS ONE’s publication criteria as it currently stands. Therefore, we invite you to submit a revised version of the manuscript that addresses the points raised during the review process.

We look forward to receiving your revised manuscript.

Kind regards,

Talib Al-Ameri, Ph.D

Academic Editor

PLOS ONE

Journal Requirements:

"Wiebke Semper-Hogg is a consultant of Dürr Dental SE.

https://www.duerrdental.com/en/home/

The study was partly funded by Dürr Dental SE.

A grant number does not exist.".

i) Please provide an amended statement that declares *all* the funding or sources of support (whether external or internal to your organization) received during this study, as detailed online in our guide for authors at http://journals.plos.org/plosone/s/submit-now.  Please also include the statement “There was no additional external funding received for this study.” in your updated Funding Statement.

ii) Please include your amended Funding Statement within your cover letter. We will change the online submission form on your behalf.

3.Thank you for stating the following in the Competing Interests section: 

"I have read the journal's policy and the authors of this manuscript have the following competing interests:

Wiebke Semper-Hogg is a consultant of Dürr Dental SE.".

 i) Please confirm that this does not alter your adherence to all PLOS ONE policies on sharing data and materials, by including the following statement: "This does not alter our adherence to  PLOS ONE policies on sharing data and materials.” (as detailed online in our guide for authors http://journals.plos.org/plosone/s/competing-interests).  If there are restrictions on sharing of data and/or materials, please state these. Please note that we cannot proceed with consideration of your article until this information has been declared.

ii) Please include your updated Competing Interests statement in your cover letter; we will change the online submission form on your behalf.

Reviewers' comments:

Reviewer's Responses to Questions

**Comments to the Author**

1. Is the manuscript technically sound, and do the data support the conclusions?

Reviewer #1: Partly

Reviewer #2: Yes

Reviewer #3: Partly

2. Has the statistical analysis been performed appropriately and rigorously? 

Reviewer #1: Yes

Reviewer #2: I Don't Know

Reviewer #3: Yes

3. Have the authors made all data underlying the findings in their manuscript fully available?

Reviewer #1: Yes

Reviewer #2: Yes

Reviewer #3: Yes

4. Is the manuscript presented in an intelligible fashion and written in standard English?

Reviewer #1: Yes

Reviewer #2: No

Reviewer #3: Yes

5. Review Comments to the Author

Reviewer #1: Thank you for submitting your work.

Summary:

In this work, the authors evaluated the doses of different CBCT devices using head and neck phantom. They found that significant dose differences between the devices could be detected.

Comments to the authors

Aim of the study:

The last paragraph in the introduction is ‘’the purpose of this study was to do further research on this topic, as side differences seem to be expectable owing to the applicational and structural differences between the devices on the market’’. Then, the authors wrote in the second paragraph in the discussion ‘’ The aim of this study was to examine side differences in distribution of dose’.

The aim was not precisely described in this manuscript, It is general and the authors have to be more specific.

methodology

The authors evaluated the ‘’side differences’’, however, they did not clarify to the readers what is/are the side difference/s, moreover, in the methodology they did not describe how they measured the side differences. The authors should describe in details the side differences and the method which they applied to measure those differences.

results

The authors did describe the distribution of dose in the result section. They should point to the figure that describes the distribution of dose.

Reviewer #2: Dear authors

Warm Greetings,

In the Discussion section : please answer the questions and do the needed corrections listed below:

1- you have to start the section by the importance and the clinical consideration of conducting your study following by mentioning the aim of the study.

2- you have to explain the causes of using each material and method items like this type of phantom, these types of CBCT devices, 3D dosimeter and organs (parotid, eye lens, thyroid and neurocranium).

3- what is your explanation about achieving the highest mean organ dose by Morita 3D Accuitomo 170?

4- side differences in distribution of dose for the parotid and eye lens were confirmed by previous research please mention them and put a reference on your explanation of that difference.

5- please clarify the explanation of side difference in a full rotation mode ( New Tom VGi evo) and mention the other devices with similar findings not only example.

6- of the eight lines of explanation on the first paragraph at the second page of the discussion section (page 9), please add a references for that explanation and rewrite it again to be more clarify.

7- At the end of the above paragraph you mention that " in literature only few studies...) please add more references and give the clinical consideration about this type of studies especially in dentistry.

8- fifth line in the second paragraph at the second page of the discussion section (page 9) to the end of the paragraph you write a lot of information without references please add references and relate with your findings.

9- the six lines of the third paragraph in the discussion section (page 9) have no references please add references.

10- line No seven in the third paragraph of the discussion section (page 9) refer to the cause of high dose to the parotid glans this is repeated information as it mentioned in the previous paragraph therefore, please rewrite it in a concise manner.

11- in the fourth paragraph of the discussion section (page 9) you mention only one old reference [10] please clarify how the finding in this reference relate to your work

12 some spelling errors at the sixth line in the second paragraph of the discussion please correct (und) between parotid and lens. another error in the 7th line of the second paragraph in page 9 (parotis)

13 please rewrite the discussion section because there is lack of flow and organization in the text .

14 lastly please use constrained statements to explain the aim of study, limitation and clinical consideration.

GOOD LUCK

Reviewer #3: Q1what's you think this work add to the subject area compared with other published work?.

Q2 Some CBCT in your research don't have high dose setting ?how manage this missing data ? for comparison purposes?.

Q3 In the abstract ,conclusion section the last row (further studies should be performed to confirm these result ) this is your conclusion or suggestion?.

Q4 In the introduction section(the main drawback of cross sectional imaging is the higher radiation dose compered to radiography ) which type of radiography?.

Q5 In method , dose measurement section can you descried in details standard dose setting and high dose setting?.

6. PLOS authors have the option to publish the peer review history of their article (what does this mean?). If published, this will include your full peer review and any attached files.

Reviewer #1: No

Reviewer #2: No

Reviewer #3: No

---

## [Author Response · Author response to Decision Letter 0]

7 May 2021

An updated Competing Interests statement was included in my cover letter.

A file responding to every reviewer´s and editor´s comment was attached in my files named "Response to Reviewers".

---

## [Decision Letter · Decision Letter 1]

26 May 2021

PONE-D-20-34852R1

Evaluation of skin doses for Cone-Beam Computed Tomography in Dentomaxillofacial Imaging - A preclinical study.

PLOS ONE

Dear Dr. Reidelbach,

Thank you for submitting your manuscript to PLOS ONE. After careful consideration, we feel that it has merit but does not fully meet PLOS ONE’s publication criteria as it currently stands. Therefore, we invite you to submit a revised version of the manuscript that addresses the points raised during the review process.

We look forward to receiving your revised manuscript.

Kind regards,

Talib Al-Ameri, Ph.D

Academic Editor

PLOS ONE

Journal Requirements:

Reviewers' comments:

Reviewer's Responses to Questions

**Comments to the Author**

1. If the authors have adequately addressed your comments raised in a previous round of review and you feel that this manuscript is now acceptable for publication, you may indicate that here to bypass the “Comments to the Author” section, enter your conflict of interest statement in the “Confidential to Editor” section, and submit your "Accept" recommendation.

Reviewer #1: All comments have been addressed

Reviewer #3: All comments have been addressed

2. Is the manuscript technically sound, and do the data support the conclusions?

Reviewer #1: Yes

Reviewer #3: Yes

3. Has the statistical analysis been performed appropriately and rigorously? 

Reviewer #1: Yes

Reviewer #3: Yes

4. Have the authors made all data underlying the findings in their manuscript fully available?

Reviewer #1: Yes

Reviewer #3: Yes

5. Is the manuscript presented in an intelligible fashion and written in standard English?

Reviewer #1: Yes

Reviewer #3: Yes

6. Review Comments to the Author

Reviewer #1: Thank you for submitting the revised version.

The authors adequately answered the reviewer comments.

I have the following comments that can improve the quality of this work:

- Page 5, last paragraph in the introduction , I would suggest the aim of the study would be as the following:

‘’The purpose of this study was to investigate the significant dose differences between the left and the right sided organs and the differences among six CBCT devices.’’

- There are some grammatical and formatting errors, proofreading is recommended for this manuscript.

Reviewer #3: (No Response)

7. PLOS authors have the option to publish the peer review history of their article (what does this mean?). If published, this will include your full peer review and any attached files.

Reviewer #1: No

Reviewer #3: No

---

## [Author Response · Author response to Decision Letter 1]

23 Jun 2021

See uploaded file "Response to Reviewers".

---

## [Decision Letter · Decision Letter 2]

29 Jun 2021

Evaluation of skin doses for Cone-Beam Computed Tomography in Dentomaxillofacial Imaging - A preclinical study.

PONE-D-20-34852R2

Dear Dr. Reidelbach,

We’re pleased to inform you that your manuscript has been judged scientifically suitable for publication and will be formally accepted for publication once it meets all outstanding technical requirements.

Kind regards,

Talib Al-Ameri, Ph.D

Academic Editor

PLOS ONE

Additional Editor Comments (optional):

Reviewers' comments:

Reviewer's Responses to Questions

**Comments to the Author**

1. If the authors have adequately addressed your comments raised in a previous round of review and you feel that this manuscript is now acceptable for publication, you may indicate that here to bypass the “Comments to the Author” section, enter your conflict of interest statement in the “Confidential to Editor” section, and submit your "Accept" recommendation.

Reviewer #1: All comments have been addressed

2. Is the manuscript technically sound, and do the data support the conclusions?

Reviewer #1: Yes

3. Has the statistical analysis been performed appropriately and rigorously? 

Reviewer #1: Yes

4. Have the authors made all data underlying the findings in their manuscript fully available?

Reviewer #1: Yes

5. Is the manuscript presented in an intelligible fashion and written in standard English?

Reviewer #1: Yes

6. Review Comments to the Author

Reviewer #1: (No Response)

7. PLOS authors have the option to publish the peer review history of their article (what does this mean?). If published, this will include your full peer review and any attached files.

Reviewer #1: No

---

## [Editor Report · Acceptance letter]

2 Jul 2021

PONE-D-20-34852R2 

Evaluation of skin doses for Cone-Beam Computed Tomography in Dentomaxillofacial Imaging - A preclinical study. 

Dear Dr. Reidelbach:

I'm pleased to inform you that your manuscript has been deemed suitable for publication in PLOS ONE. Congratulations! Your manuscript is now with our production department. 

Kind regards, 

on behalf of

Dr. Talib Al-Ameri 

Academic Editor

PLOS ONE